# Adsorption of phosphate over a novel magnesium-loaded sludge-based biochar

**Chu-Ya Wang**[ID]*, **Qi Wang, Heng-Deng Zhou, Xin Fang, Qi Zeng, Guangcan Zhu**[ID]

School of Energy and Environment, Southeast University, Nanjing, China

\* wang-cy@seu.edu.cn

## Abstract

The production of sludge-based biochar to recover phosphorus (P) from wastewater and reuse the recovered phosphorus as agricultural fertilizer is a preferred process. This article mainly studied the removal of phosphate ($PO_4$-P) from aqueous solution by synthesizing sludge-based biochar (MgSBC-0.1) from anaerobic fermentation sludge treated with magnesium (Mg)-loading-modification, and compared it with unmodified sludge-based biochar (SBC). The physicochemical properties, adsorption efficiency, and adsorption mechanism of MgSBC-0.1 were studied. The results showed that the surface area of MgSBC-0.1 synthesized increased by 5.57 times. The material surface contained MgO, $Mg(OH)_2$, and CaO nanoparticles. MgSBC-0.1 can effectively remove phosphate in the initial solution pH range of 3.00–7.00, with a fitted maximum phosphorus adsorption capacity of 379.52 mg·g$^{-1}$. The adsorption conforms to the pseudo second-order kinetics model and Langmuir isotherm adsorption curve. The characterization of the adsorbed composite material revealed the contribution of phosphorus crystal deposition and electrostatic attraction to phosphorus absorption.

## Introduction

In recent years, with the acceleration of the industrialization process, the scale of sewage treatment plant (WWTPs) has gradually increased, as has the production of sludge as a by-product of WWTPs [1, 2]. Phosphorus in sewage is the main source of excessive phosphorus discharge in water bodies. Excessive discharge of phosphorus containing wastewater can lead to eutrophication of surface water, and sludge is composed of organic matter, trace elements, microorganisms [3]. and other toxic substances such as pathogens and heavy metals [4, 5]. Therefore, strengthening sludge treatment is of great significance in addressing environmental pollution risks and phosphorus crises. The difference between sludge pyrolysis and traditional compost landfill lies in its strong stability, fast processing speed, and non-toxic product. This is one of the most promising processes for high-value utilization of biomass [6]. Most notably, the produced sludge-based biochar can be utilized as an adsorbent to treat fuel, phenols, and phenolic chemicals in waste water since it is a porous material with enough surface functional groups [7, 8], therefore, the use of biomass waste to produce new low-cost green adsorbents has gradually attracted attention to green development and carbon neutrality. In recent years,

**Data Availability Statement:** All relevant data are within the manuscript and its Supporting Information files.

**Funding:** This study was supported by Chu-Ya Wang from the Natural Science Foundation of

Jiangsu Province (BK20211047) and Guangcan Zhu from the Natural Science Foundation of Jiangsu Province(BK20220038). Chu-Ya Wang: Conceptualization and Funding acquisition Guangcan Zhu:Funding acquisition and Validation.

**Competing interests:** The authors have declared that no competing interests exist.

substantial progress has been made in the preparation of sludge-based biochar using sludge as raw material through chemical modification.

The adsorption capacity and efficiency of unmodified BC are not high enough, and modification treatment is needed. The commonly used modification methods include electrochemical modification, biomass pyrolysis, and bioaccumulation [9]. The most typical way was to alter biochar (BC) with metal ions using various methods such as prediction. The aluminum/magnesium (Al/Mg) modified BC can achieve a phosphate adsorption capacity of 74.47 mg·g$^{-1}$ by employing an Al electrode and a concentrated Mg solution as the electrolyte [10]. Other metals, such as calcium (Ca) and Mg, are also employed to increase phosphate adsorption by BC. Before pyrolysis, BC raw materials can be mixed with materials rich in Ca and Mg ions, such as dolomite and montmorillonite. Mg/Ca modified biochar has a higher capacity for Mg and P adsorption [11, 12]. When sugarcane straw biochar is changed with Al, it may adsorb phosphate up to 758.96 mg·g$^{-1}$; when modified with Mg/Al, it can adsorb up to 887 mgP·g$^{-1}$ [13]. This is because Al and Mg can add new adsorption sites and undergo chemical and physical adsorption simultaneously.

SBC has the advantage of low cost and is one of the best adsorbents for phosphorus treatment in wastewater treatment plants [1]. And SBC can be reused in sewage treatment plants for phosphate or other pollutant adsorption treatment. However, there is limited research on the use of SBC for P adsorption, possibly due to its low adsorption capacity for phosphate in wastewater, so modification of SBC is necessary. At present, there are modification methods such as inorganic salt treatment, inorganic acid-base treatment [14], activation [15], and other biomass impurities. Among them, metal impurity treatment is the most widely used, such as Al, Fe, Lanthanum (La) and other impurities, but it can cause further pollution, and the modification object is generally activated carbon or wooden biochar [10, 16, 17]. According to reports in recent years, adsorbents treated with Mg salts exhibit strong phosphate activity and affinity, potentially enhancing their adsorption capability [18–20]. However, the effectiveness of Mg salt modified sludge-based biochar in adsorbing phosphate is still unclear, and there is also limited research on the desorption mechanism and release characteristics of phosphate.

In this study, a series of sludge-based biochar material with different Mg-loading-modification were synthesized by impregnation method. The morphology, pore structure, surface chemical state, and distribution and existing form of Mg in the prepared Mg-loaded sludge-based biochar were comprehensively characterized. Determine the mechanism for controlling phosphorus adsorption and measure the adsorption characteristics of sludge-based biochar on phosphorus through kinetics and isothermal adsorption experiments. Additionally, studies on the impact of coexisting ions on the adsorption of phosphate on sludge-based biochar helped to clarify the molecular mechanism of the generated material's enhancement of phosphorus adsorption. More importantly, the effectiveness and repeatability of phosphorus recovery had been confirmed, which showed great potential for the practical application of these as-prepared sludge-biochar.

## Materials and method

### Preparation of sludge-biochar

Anaerobic fermentation sludge from the sludge process served as the raw material for the production of biochar. The sludge was dried in a 105°C oven to reduce water to less than 1%, ground to 200 mesh.

A 0.1 mol·L$^{-1}$ Mg (CH$_3$COO)$_2$ solution was prepared, from which a 100 mL sample was taken in a baker, and 2 g sludge powder (SP) was added. The mixture was stirred at a constant rate at room temperature for 6 h, and then the beaker was placed in the oven for 24 h to

remove remaining water. The dried material was removed from the beaker and then ground until it passed through a 200 mesh sieve. The material after grinding was named MgSP, which was calcined in a tube furnace under a helium atmosphere at 500°C for 2 h at a heating rate of 10°C·min$^{-1}$. After cooling, the material was removed and named MgSBC-0.1. All of the reagents used in the experiment were acquired from Aladdin Industrial Company in Shanghai, China. Prepare a phosphate solution (1000 mg·L$^{-1}$) from the assessed and graded potassium dihydrogen phosphate ($K_2HPO_4$) in deionized water and dilute it to the appropriate concentration for further batch studies.

## Batch adsorption experiments

To better understand the adsorbing effect of biochar on phosphates, $KH_2PO_4$ was chosen to represent inorganic phosphate.

The bath adsorption method was used to examine the effects of initial pH, initial concentration of the phosphate solution, amount of adsorbent employed, and contact time. The dosage of adsorbent was fixed at 0.67 g·L$^{-1}$. For the contact time test, the solution-filled vial was shaken at 300 rpm for one to seventy-two minutes at room temperature.

Calculate the amount of phosphate adsorbed at that time using Eq (1):

$$Q_t = \frac{V}{M}(C_0 - C_t) \tag{1}$$

Where $Q_t$ refers to the amount of phosphorus adsorbed by each unit of adsorbent at time t. Where $C_0$ (mg·L$^{-1}$) represents the initial phosphorus concentration, $C_t$ (mg·L$^{-1}$) represents the phosphorus concentration at time t, and ($C_0$-$C_t$) refers to the change of phosphorus concentration. The mass of the adsorbent and the volume of the solution are represented by $M$ and $V$, respectively.

Without pH adjustment, the pH value of 100 mg L$^{-1}$ $KH_2PO_4$ solution is around 5.60. Using the same method as the contact time test, but with different dosages. 0.67 g·L$^{-1}$ of adsorbent was used to test the adsorbent dose study. The first pH investigation was then carried out by adjusting the initial pH values of the phosphate solution from 3 to 11 using HCl or NaOH.

In isothermal adsorption studies, dilute phosphate solution to various concentrations (0, 5, 10, 20, 40, 50, 100, 200, 250, and 400 mg·L$^{-1}$). Adjusting the reaction temperature (to 25, 35, and 45°C) was how the experiment was carried out, and the fixed adsorbent dosage was 0.67 g·L$^{-1}$.

Following the equilibrium of the reaction, the concentration of phosphate left in the solution could be determined. Then, using Eq (2), the phosphate's adsorption capacity ($Q_e$, mg·g$^{-1}$) could be computed.

$$Q_e = \frac{V}{M}(C_0 - C_e) \tag{2}$$

where $C_e$ (mg·L$^{-1}$) is the phosphate concentration at equilibrium.

Add common anions Cl$^-$, NO$_3^-$, SO$_4^{2-}$, CO$_3^{2-}$, and HCO$_3^-$, as well as cations NH$_4^+$, Fe$^{3+}$, and Ca$^{2+}$ that may appear in wastewater, to 0.01, 0.05, and 0.10 mmol·L$^{-1}$ phosphate solutions, respectively, to prepare binary solutions and investigate the competitive performance of anions and cations.

$$D = \frac{Q_e - Q_a}{Q_e} \times 100\% \tag{3}$$

In the presence of competing ions, the adsorbent's adsorption capacity on phosphate is expressed as $Q_a$ (mg·g$^{-1}$).

## Characterization and analytical method

Mo-SS anti-spectrophotometry was used, and a spectrophotometer (752 N-116 spectrophotometer; China). A pH meter (pHS-3C, China) was used to determine the solution's pH value. The Quantachrome AUTOSORB IQ from the United States was used to evaluate the adsorption desorption isotherm of $N_2$. Using the Barrett Joyner Helenda (BJH) method, determine the specific surface area ($S_{BET}$) and other pore parameters of MgSBC-0.1. Using X-ray diffraction (XRD), Fourier Transform Infrared Spectroscopy (FTIR), (X-Ray Photoelectron Spectroscopy) XPS, and SEM, the differences between MgSBC-0.1 and P adsorbed MgSBC-0.1 were examined. Capture XRD spectra within the region of 10~80˚ using a Cu K α radiation diffractometer (utima IV) (λ = 1.5418 a) at 2θ. Then, analyze the spectra using MDI Jade 6 software. 2˚ min$^{-1}$. Pure KBr should be ground finely, then pressed into transparent thin sheets using an oil press. Transmission electron microscope (TEM) produced by Thermo Fisher Scientific, USA (Talos F200X). The sample should be characterized using FTIR on Nicolet IS10 catalyst with a wave number range of 4000–400 cm$^{-1}$. The sample should be placed in the sample chamber of the Thermo Scientific K-Alpha XPS instrument, where the excitation source is Al K α Radiation (hv = 1486.6 eV) and the sample chamber pressure should be less than 2.0 × Perform XPS detection at $10^{-7}$ mbar.

## Results and discussion

### Characterization and morphology

In order to observe the changes in the pore structure of biochar after adding magnesium and calcination, SEM testing was conducted. The morphology of SP, SBC, MgSP-0.1, and MgSBC-0.1 is shown in the Fig 1. SP and SBC have a granular structure covered by pores, with shallow pore structures (Fig 1A and 1B). These pores and pore structures were irregular. The sludge powder impregnated with magnesium acetate has a smooth surface, slender pores, and a linear shape (Fig 1C). Through Mg-loading and calcination modification, the pores of MgSBC-0.1

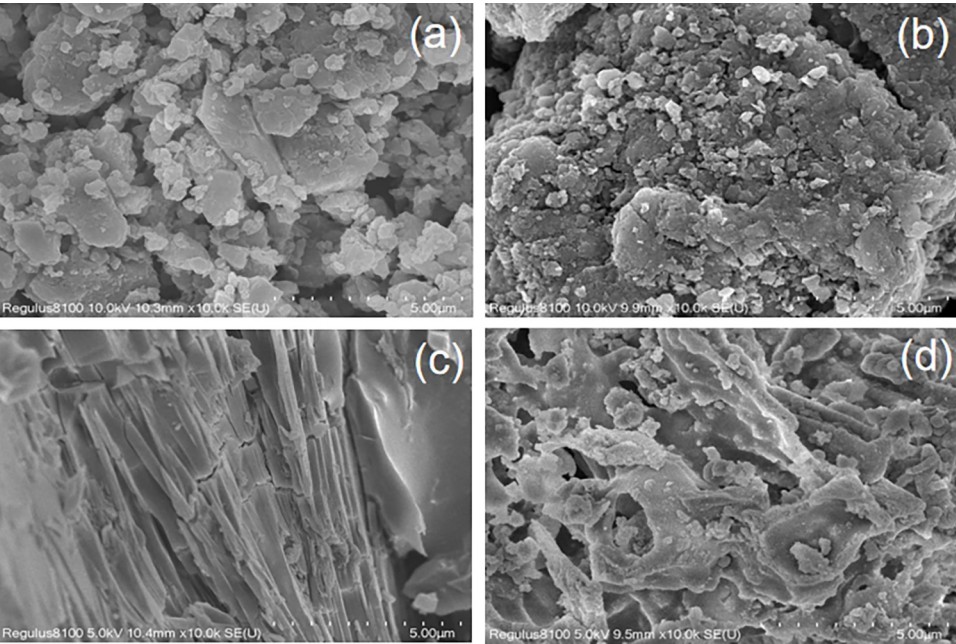

**Fig 1.** SEM image of (a) Original dry sludge, (b) SBC, (c) MgSP-0.1 and (d) MgSBC-0.1, respectively.

are stretched, with some pore sizes increasing and pore depths deepening, providing additional surface area for adsorption and increasing adsorption sites (Fig 1D).

In order to ascertain the adsorbent's surface area and pore structure, we developed an isotherm for nitrogen adsorption desorption at a constant temperature of 77 K. BET testing was conducted on the material. The BET adsorption and desorption curves of four materials are shown in Fig 2A. In terms of BET surface area ($cm^2 \cdot g^{-1}$) and pore volume ($cm^3 \cdot g^{-1}$), SBC had values of 17.76 and 0.037, while MgSBC-0.1 had values of 93.35 and 0.17. The surface area and pores of SBC were substantially less than those of MgSBC-0.1, according to this result. The variations in particular surface area matched the findings of the SEM. Moreover, the average pore size (APD) of SP, SBC, MgSP-0.1, and MgSBC-0.1 were 12.92, 10.27, 4.19, and 6.3860 nm, respectively. During the modification process, the average pore sizes of SP, SBC, and MgSBC-0.1 were still between 2–50 nm, all belonging to mesopores. However, it may be due to the blockage of some pores by MgO impregnated biochar and the high calcination temperature, causing some pores of MgSBC-0.1 to collapse during the preparation process, resulting in a lower APD. The findings showed that biochar's pore shape and surface area could be modified by calcination and Mg-loading, which had an impact on the material's ability to adsorb and desorb substances. According to the IUPAC classification, the $N_2$ adsorption curves of SP, SBC, and MgSBC-0.1 showed type IV adsorption isotherms, accompanied by $H_3$ hysteresis loops. This behavior could be attributed to the presence of mesopores formed through the aggregation of plate-like particles. The materials SP, SBC, MgSP-0.1, and MgSBC-0.1 all displayed vertical adsorption branches characterized by relative pressures in proximity to unity. Both SBC and MgSBC-0.1 samples displayed hysteresis loops characterized by vertical desorption branches at a relative pressure of around 0.5. The hysteresis loop seen in this study was produced by pores that were slit-shaped, including thin, short, and parallel openings [21]. Fig 2B shows the XRD patterns of the crystal structures of SP, SBC, and MgSBC-0.1. The results indicated that the peaks corresponding to the crystals present in biochar were matched with $SiO_2$ (PDF # 85–0798) and MgO (PDF # 45–0946). The diffraction peaks obtained at $2\theta$ = 36.86˚ (111), 42.83˚ (200), 62.30˚ (220), 74.69˚ (311), and 78.63˚ (222) match MgO, while the diffraction peaks obtained at $2\theta$ = 20.86˚ (100), 26.64˚ (011), 36.54˚ (110), 39.47˚ (102), and 50.14˚ (11–2) match $SiO_2$. It could be seen that SP was mainly composed of silicon dioxide ($SiO_2$). Calcination increased the peak strength of $SiO_2$ in biochar. In addition, the peak intensity of $SiO_2$ in MgSBC-0.1 was lower than that of unmodified biochar. However, a MgO diffraction peak appeared in the XRD spectrum of MgSBC-0.1, with the strongest peak occurring at $2\theta$ = 42.83˚, indicating the presence of a large amount of MgO in the prepared biochar after impregnation and calcination, which was also confirmed by other techniques (XPS and TEM) [22].

The method of surface semi-quantitative elemental analysis X-ray photoelectron spectroscopy (XPS) was employed to investigate the elemental composition (Fig 2C). Based on the XPS analysis outcomes presented in Table 1, the surface composition of SP is found to consist of carbon (C), oxygen (O), nitrogen (N), silicon (Si), Ca, and iron (Fe). The presence of Mg in MgSBC-0.1 suggested the successful loading of Mg onto the biochar material. The surface semi quantitative elemental analysis method XPS was used to analyze the elemental composition. According to the XPS characterization results in Table 1, the surface of SP contains C, O, N, Si, Ca, and Fe. And MgSBC-0.1 contained Mg, indicating that Mg had been successfully loaded onto biochar. The O/C ratios of SP and MgSBC-0.1 were determined to be 0.50 and 1.49, respectively, through calculation. The O/C ratio served as an indirect measure of the overall concentration of surface groups containing oxygen. This implied that the cumulative concentration of oxygen-containing functional groups (OFGs) on the surface. The electrical conductivity of MgSBC-0.1 was almost three times greater than that of SP. The OFG present on

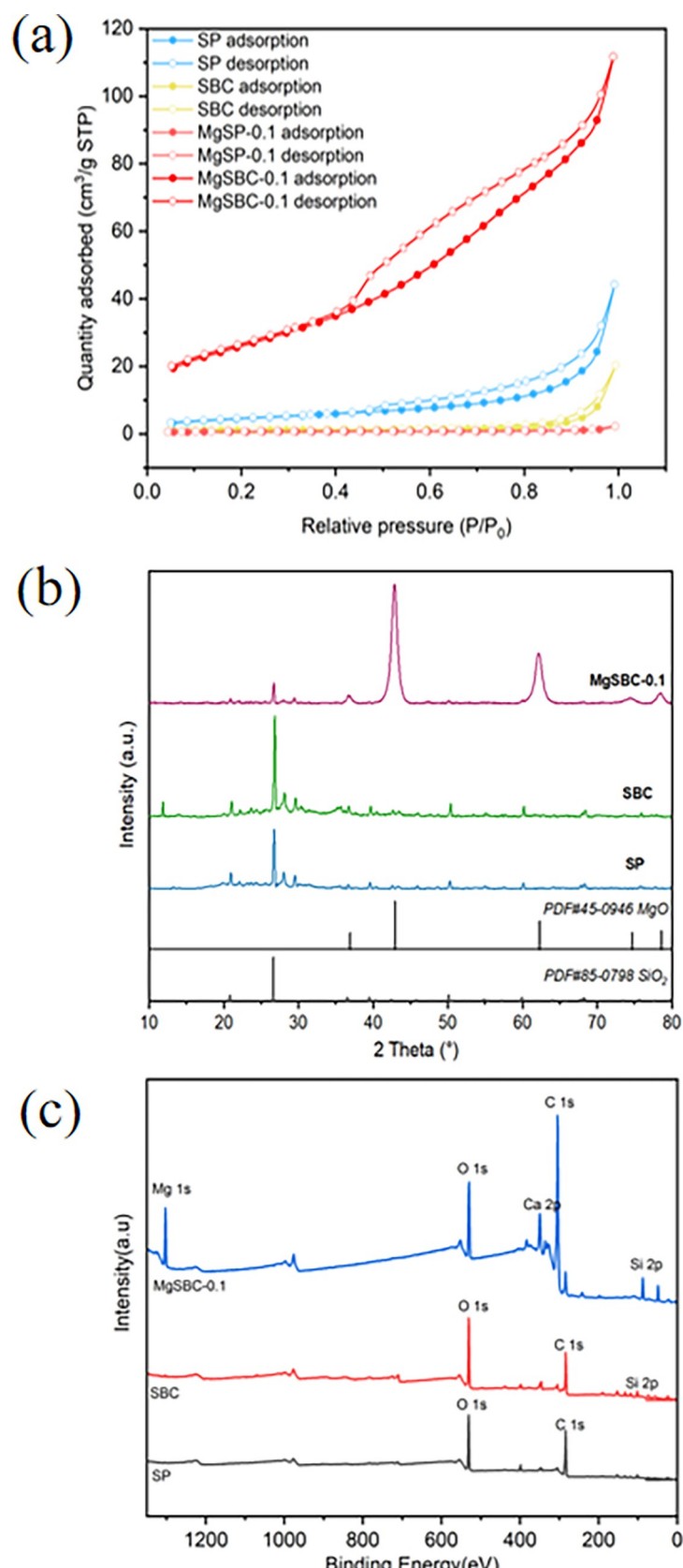

**Fig 2.** (a) $N_2$ adsorption and desorption curve, (b) XRD patterns of SP, SBC and MgSBC-0.1 and (c) XPS spectra of SP, SBC, and MgSBC-0.1.

biochar played a crucial role as active sites in the process of removing pollutants from water, mostly by interfacial adsorption and redox reactions. This phenomenon held potential benefits for the removal of phosphates from water sources [23].

The distribution of Fe elements within the MgSBC-0.1 material was observed to occur at various depths, rather than restricting solely to the surface. During the process of adsorption, the internal iron element underwent dissolution in the solution, subsequently forming a compound with P and precipitating onto the surface. This interaction enhanced the adsorption of phosphorus [24]. The chemical reaction between the Ca presented in the sludge and phosphate could enhance the adsorption capacity of MgSBC-0.1 [25].

The micro morphology of SP, SBC, MgSBC-0.1 is visualized by TEM images (Fig 3A–3D). SP and SBC have no obvious pore structure at 500 nm, but the morphology of mgsbc-0.1 was different from this. It can be clearly seen from Fig 3D that MgSBC-0.1 produces more pores, indicating that more active sites may be exposed on the surface. The HRTEM image at 5 nm shows the d spacing between crystal planes of 0.209 nm (Fig 3E), corresponding to (200) of MgO, which may indicate that MgO is loaded on the surface of biochar, but it needs further verification. The element distribution in the mapping diagram of EDS further confirmed the element distribution in MgSBC-0.1. As shown in Fig 3F–3I, Mg, O and Si elements are well dispersed in the whole particle.

Fig 4 illustrates the FTIR spectra of SP, SBC, and MgSBC-0.1 within the wavenumber range of 4000–400 $cm^{-1}$. The common deformation vibration absorption peaks of SP, SBC, and MgSBC-0.1 were 802, 1086, 1624, 2918 and 3427 $cm^{-1}$, respectively. The observed peaks in the vibrational absorption spectra could be attributed to various molecular vibrations. Specifically, the peaks corresponded to the deformation vibrations of aromatic compound C-H bonds, the stretching vibrations of C-O bonds, the symmetric and asymmetric stretching vibrations of aliphatic or cycloalkane -$CH_3$ and -$CH_2$ groups, and the stretching vibrations of -OH groups in intermolecular hydrogen bonded alcohols and phenols [26, 27]. This observation suggested that the fundamental characteristics of biochar remained mostly unchanged during the MgSBC-0.1 alteration procedure. Furthermore, it was observed that MgSBC-0.1 exhibited vibrations at a wavenumber of 3745 $cm^{-1}$, indicating a correlation with the formation of Mg-OH bonds. This finding suggested that the presence of biochar resulted in the production of $Mg(OH)_2$, which enhanced the adsorption of phosphate.

## Phosphate adsorption isotherm and kinetics

Using a 100 mg·$L^{-1}$ P solution, 2 g of sludge powder was immersed in 0.01, 0.05, 0.1, and 0.2 mol of Mg acetate solution (named MgSBC-0.01, MgSBC-0.05, MgSBC-0.1, and MgSBC-0.2) to compare the adsorption performance of four different Mg-loaded sludge-based biochar materials for phosphate. According to the findings presented in Fig 5, it is evident that the

**Table 1. Elemental composition (Atomic%) and oxygen carbon atomic ratio of SP, SBC, P-MgSBC-0.1 and Des-MgSBC-0.1 estimated by X-ray photoelectron spectroscopy (XPS).**

| Sample | C | O | Si | Mg | N | Ca | Fe | O/C |
|---|---|---|---|---|---|---|---|---|
| SP | 59.02 | 29.81 | 3.65 | / | 5.34 | 1.22 | 0.96 | 0.51 |
| SBC | 48.18 | 35.34 | 6.49 | / | 5.08 | 2.55 | 2.36 | 0.73 |
| MgSBC-0.1 | 28.44 | 42.48 | 1.48 | 14.46 | 2.57 | 10.40 | 0.17 | 1.49 |

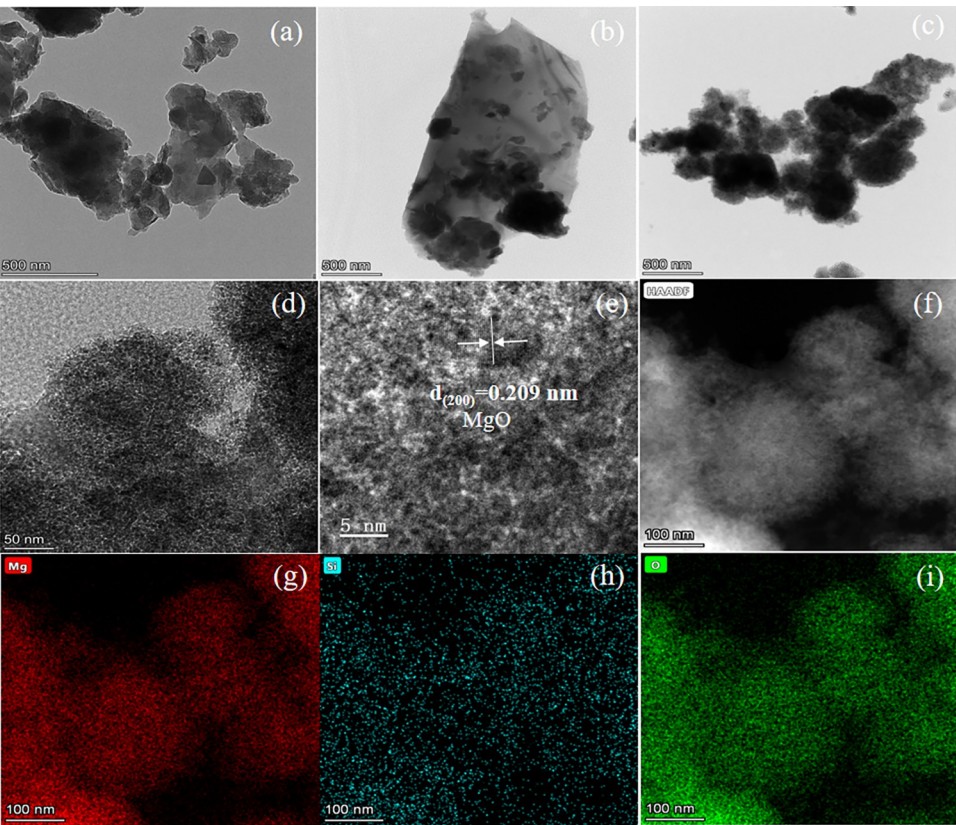

**Fig 3.** (a-c) TEM images (500nm) of SP, SBC and MgSBC-0.1, (d-e) TEM images of MgSBC-0.1, (f-i) Elemental mapping of MgSBC-0.1.

incorporation of Mg-loading-modification into carbon resulted in a discernible capacity for phosphorus removal. Moreover, the introduction of Mg oxide/hydroxide loading further enhanced the adsorption efficacy of sludge carbon in relation to phosphorus. The adsorption capacity of sludge-based biochar for P initially increased and then reached a state of progressive stabilization as the loading of Mg increases. At this point, the adsorption capacity reached approximately 140 mg·g$^{-1}$, suggesting that there existed a specific threshold for the loading capacity of sludge powder. In order to delve deeper into the process of phosphorus removal, an adsorption isotherm was generated with MgSBC-0.1.

To enhance comprehension of the adsorption characteristics of MgSBC-0.1 on phosphate, the experimental isotherm was simulated using the Langmuir and Freundlich isotherm equations, as shown below.

$$Q_e = Q_{max} K_L C_e / (1 + K_L C_e) \tag{4}$$

$$Q_e = K_f C_e^{1/m} \tag{5}$$

where $Q_{max}$ is the maximum adsorption capacity of the adsorbent (mg·g$^{-1}$), $K_F$ [mg (g·(L·mg$^{-1}$)$^{1/n}$)$^{-1}$] and $K_L$ (L mg$^{-1}$) are constants of the Freundlich and Langmuir model equations, respectively. The observed trend in Fig 6A and 6B indicates that the phosphate adsorption capacity exhibited a substantial rise with the rising concentration of phosphoric acid, eventually reaching a state of equilibrium. The results of fitting the experimental data indicated that the Langmuir model and Freundlich model achieved $R^2$ values of 0.94 and 0.85, respectively,

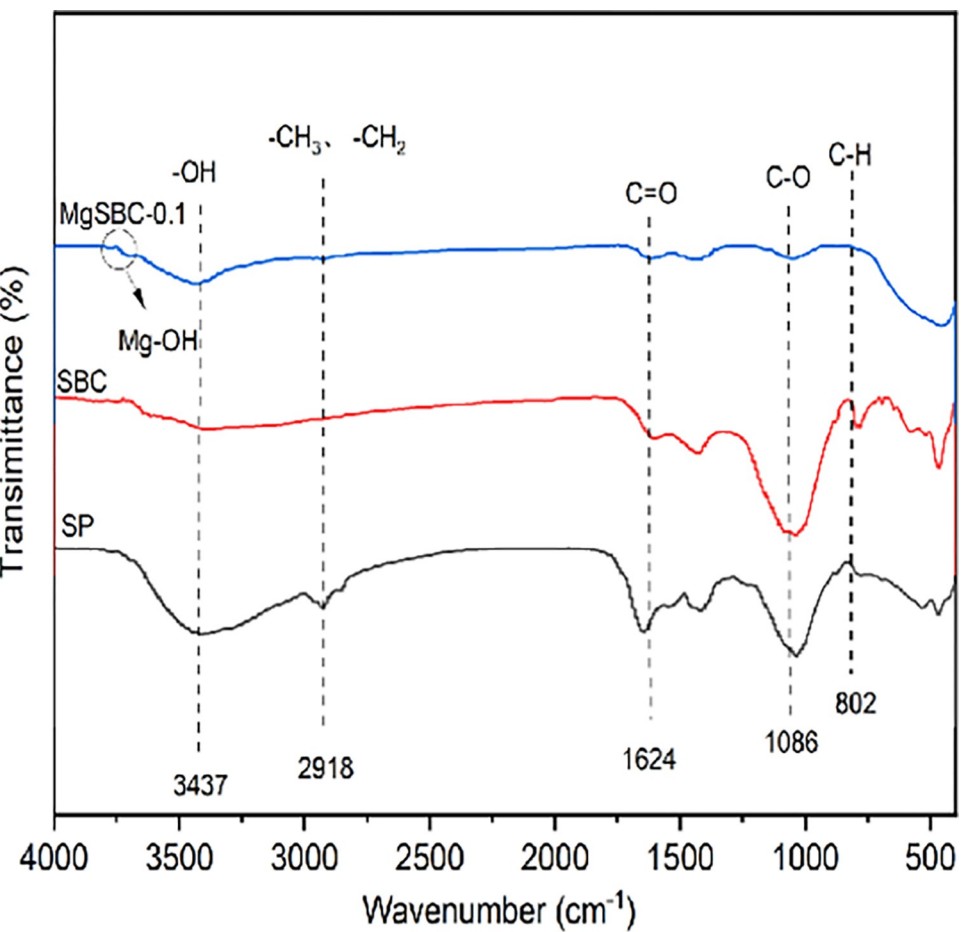

**Fig 4. FTIR spectra of SP, SBC, and MgSBC-0.1.**

at a temperature of 298 K (as shown in Table 2). The Langmuir model, which was found to be a good fit, demonstrated the homogenous adsorption of phosphorus on MgSBC-0.1. This adsorption process was determined to be a monolayer adsorption. Furthermore, the phenomenon could be elucidated by the process of phosphate undergoing surface crystallization to form Mg compounds [24, 28, 29]. The Langmuir model was utilized to determine the maximal phosphate adsorption capacity, yielding a value of 379.52 mg·g$^{-1}$. In comparison to unaltered biochar, which exhibited an adsorption capacity of 52.95 mg·g$^{-1}$ (S1 Fig), MgSBC-0.1 substantially increased its phosphate adsorption capacity by 7.17 times. However, the surface area of MgSBC-0.1 increased by 5.57 times higher than that of SBC. Hence, the notable increase in phosphate adsorption capacity could be ascribed to the chemical adsorption process facilitated by the presence of MgO on the biochar surface. But obviously, the increase in specific surface area is not the only factor causing the increase in adsorption capacity of MgSBC-0.1.

To facilitate a more comprehensive examination of the adsorption characteristics of MgSBC-0.1 on phosphate, various kinetic models including pseudo first order, pseudo second order, and intra-particle diffusion models were employed for analysis. Fig 6C illustrates the adsorption capability of MgSBC-0.1 for P. The MgSBC-0.1 material exhibited a phosphate capture efficiency of almost 70 mg·g$^{-1}$ within a 15 min timeframe. Equilibrium was achieved in roughly 2 h, with the adsorption capacity being stable at around 150 mg·g$^{-1}$. The rapid capture

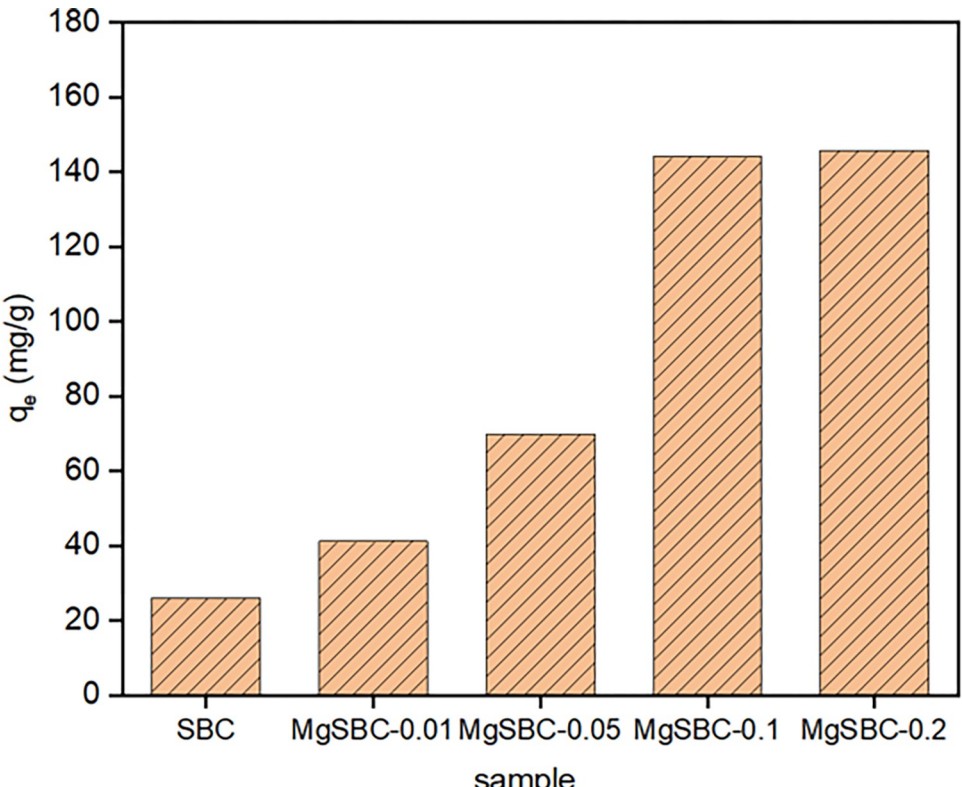

**Fig 5. The effect of magnesium loading on the adsorption of P by Mg-loaded sludge-biochar.**

of phosphate ions could be attributed to the strong electrostatic attraction between the positively charged MgSBC-0.1 and the negatively charged phosphate ions present on the surface.

The phosphate adsorption process of MgSBC-0.1 was described using both the pseudo-first-order model and the pseudo-second-order model.

$$\ln(Q_e - Q_t) = \ln Q_e - k_1 t \tag{6}$$

$$t/Q_t = 1/(k_2 Q_e^2) + t/Q_e \tag{7}$$

where $k_1$ (min$^{-1}$) and $k_2$ (g (mg·min)$^{-1}$) are the adsorption rate constants of the pseudo-first-order model and the pseudo-second-order model.javascript:void(0);

To enhance the understanding of the impact of the porous structure of MgSBC-0.1, the experimental data was simulated using the intra particle diffusion model.

$$Q_t = K_t^{0.5} + C \tag{8}$$

where $K$ (mg·min$^{0.5}$)·g$^{-1}$ is the intra particle diffusion rate constant, and $C$ is a constant related to the thickness of the adsorption boundary layer.

The analysis of Table 3 reveals that the correlation coefficients obtained from the pseudo second-order dynamic model simulations are superior to those obtained from the pseudo first-order dynamic model simulations. This suggested that MgSBC-0.1 exhibited a stronger compatibility with the pseudo second-order dynamic model. The pseudo second-order kinetic model postulated that the elimination mechanism of adsorbed compounds was governed by chemical adsorption [30], which included valence electron force, electron exchange or sharing

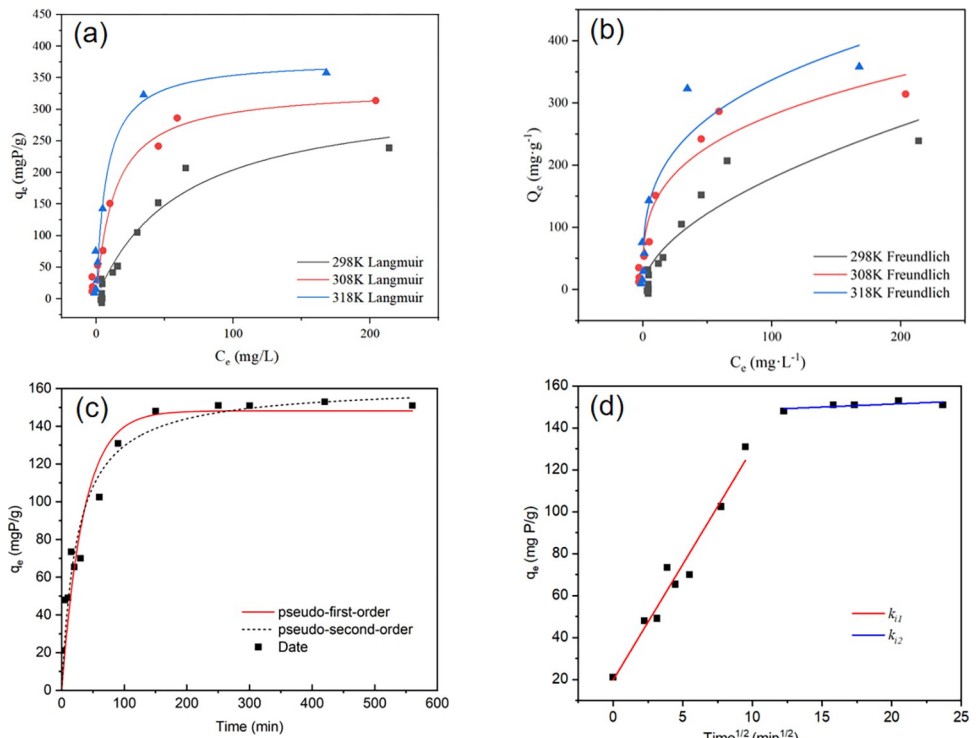

**Fig 6.** The phosphate adsorption isotherms and the kinetic curve (a and b) The fitting the phosphate adsorption isotherms of MgSBC-0.1 at different temperatures, (c) the kinetic curve of MgSBC-0.1, (d) particle diffusion model fitting of MgSBC-0.1.

between the adsorbent and the adsorbed substance, and the possible formation of new compounds [24]. As shown in S2A Fig, the kinetic curve of SBC is more in line with the pseudo first-order kinetic model. After Mg loading modification, the adsorption mechanism of phosphate exhibited chemical adsorption control, and the introduced $Mg^{2+}$ underwent a chemical reaction with $H_2PO_4^-$. This might be the fundamental reason for improving adsorption performance.

In order to further elucidated the influence of MgSBC-0.1 porous structure, the particle internal diffusion model was used to simulate the experimental data. Based on the data presented in Fig 6D, it is evident that in the two-stage intra-particle diffusion process, the value of $k_{i1}$ (22.04) surpasses that of $k_{i2}$ (0.047). This discrepancy suggested that the rate of adsorption was primarily governed by external diffusion, with intra-particle diffusion exerting minimal

**Table 2. Adsorption kinetics parameters of MgSBC-0.1.**

| Models | Parameters | Values |
|---|---|---|
| Pseudo-first-order | $q_e$ (mg P $g^{-1}$) | 148.20 |
| | $k_1$ ($min^{-1}$) | 0.03 |
| | $R^2$ | 0.90 |
| Pseudo-second-order | $q_e$ (mg P $g^{-1}$) | 162.03 |
| | $k_2$ (g (mg min)$^{-1}$) | $2.46 \times 10^{-4}$ |
| | $R^2$ | 0.93 |
| Intraparticle diffusion | $k_{i1}$ (mg $g^{-1}$ $min^{-0.5}$) | 11.04 |
| | $k_{i2}$ (mg $g^{-1}$ $min^{-0.5}$) | 0.28 |

**Table 3. Isothermal adsorption model parameters of MgSBC-0.1 at 298, 308, and 318 K, respectively.**

| Models | Parameters | Values | | |
|---|---|---|---|---|
| | | 298 K | 308 K | 318 K |
| Langmuir | $Q$ (mg·g$^{-1}$) | 329.16 | 333.92 | 379.52 |
| | $K_L$ (L·mg$^{-1}$) | 0.01635 | 0.07541 | 0.13782 |
| | $R^2$ | 0.9437 | 0.9663 | 0.99704 |
| Freundlich | $K_F$ | 14.01282 | 72.21307 | 86.0854 |
| | 1/n | 0.55306 | 0.29433 | 0.29621 |
| | $R^2$ | 0.84722 | 0.92805 | 0.85997 |

influence on the overall adsorption rate. For SBC, $k_{i1}$ (5.35) was also greater than $k_{i2}$ (0.61), indicating that the adsorption of phosphate by sludge-based biochar before and after modification is determined by external diffusion (S2B Fig). The $k_{i1}$ of MgSBC-0.1 was greater than $k_{i1}$ of SBC, indicating that enhanced external diffusion contributes more to the improvement of phosphate adsorption capacity [31].

## Effect of different factors on phosphate capture

According to the data presented in Fig 7, it can be observed that the adsorption capacity of MgSBC-0.1 for phosphate remains relatively constant within the pH range of 3–6. The maximum adsorption capacity recorded is around 140 mg·g$^{-1}$. In the specified pH range, the phosphate in the solution was mostly in the $H_2PO_4^-$ form, leading to the precipitation of the liberated $Mg^{2+}$ and subsequent formation of $MgH_2PO_4$ crystals. Furthermore, it was observed that the adsorption capacity for phosphorus reached a maximum value of 146 mg·g$^{-1}$ at a pH level of 5.00. When the pH value of the solution exceeded 6 and continues to increase, the adsorption capacity for phosphorus gradually decreased. At a pH value of 11, the adsorption capacity of phosphorus exhibited a substantial decline, reaching 46.66 mg·g$^{-1}$. This decrease amounted to a reduction of 68.04% in comparison to the adsorption capacity seen at pH 5.00. The predominant phosphate species found in the solution were the monovalent $H_2PO_4^-$ ion ($pK_1 = 2.15$) at pH values below 7.20, the divalent $HPO_4^{2-}$ anion ($pK_2 = 7.20$) at pH 7.20, and the trivalent $PO_4^{3-}$ anion at pH values above 12.33 [32, 33]. The above findings indicated that when the pH approached 7.00, a substantial increase in negative surface charge and the presence of polyvalent phosphorus oxide anions contributed to the enhancement of electrostatic repulsion [34]. Given the typical pH level of approximately 7 in wastewater, it could be inferred that the utilization of MgSBC-0.1 was appropriate for the purpose of phosphorus recovery from wastewater. Furthermore, this approach had demonstrated favorable outcomes in terms of adsorption treatment efficacy.

The composition of wastewater was complex, and the adsorption capacity of MgSBC-0.1 for phosphate was affected by various coexisting ions in the wastewater Therefore, the impact of various cations, such as $Ca^{2+}$, $Fe^{3+}$, and $NH_4^+$, and anions, such as $NO_3^-$, $Cl^-$, $SO_4^{2-}$, $CO_3^{2-}$, and $HCO_3^-$, on the phosphate adsorptive removal by MgSBC-0.1 were studied. Moreover, humic acid (HA), a common organic matter in wastewater, was also an important factor affecting the adsorption of phosphate by MgSBC-0.1 and needs to be studied. As is shown in Fig 8A, the coexistence of these five anions has varying degrees of impact on the performance of MgSBC-0.1 in order of $HCO_3^- > CO_3^{2-} > SO_4^{2-} > Cl^- > NO_3^-$. Among these five anions, $HCO_3^-$ and $CO_3^{2-}$ had the greatest competitive effect on phosphate due to their competitive binding sites with phosphate [34]. When the concentrations of anions were 0.01, 0.05, and 0.1 mmol·L$^{-1}$, the inclusion of $Cl^-$, $NO_3^-$, and $SO_4^{2-}$ ions had no substantial impact on the adsorption

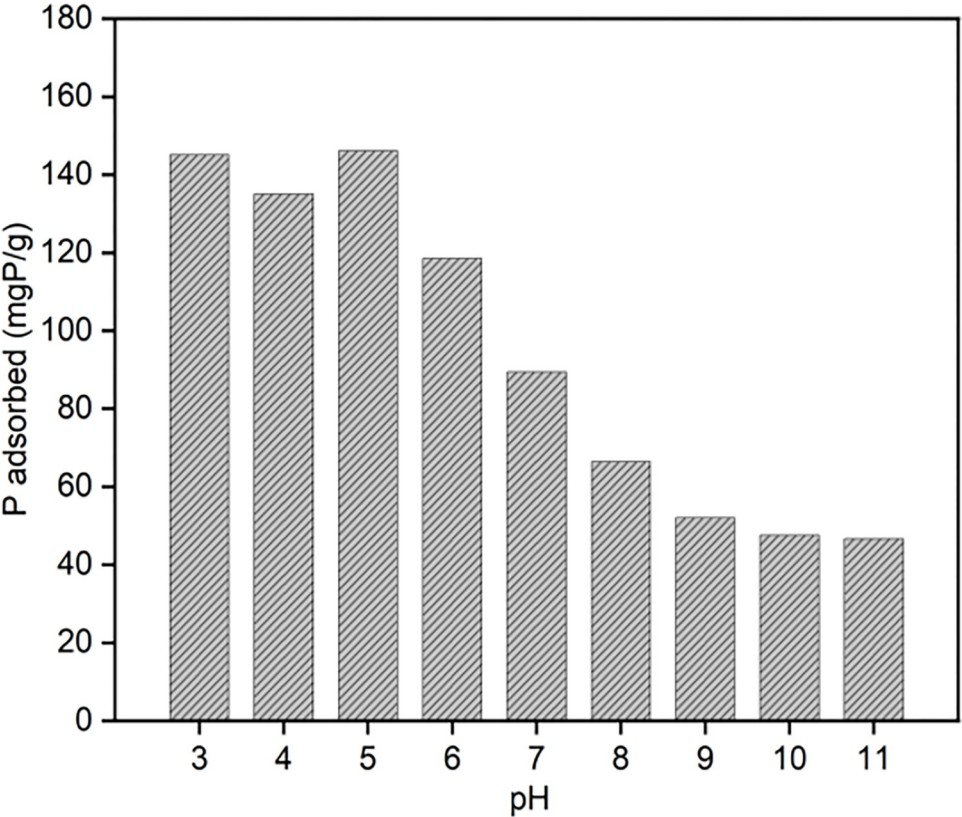

**Fig 7. The effect of initial pH on phosphate capture.**

outcomes of phosphate by MgSBC-0.1 in comparison to a pH value of 7.00 (Fig 8A). This observation provided evidence that anions such as $Cl^-$, $NO_3^-$ and $SO_4^{2-}$ were unable to undergo precipitation with $Mg^{2+}$. Instead, they tended to form outer-sphere surface complexes, resulting in minimal impact on the sorption of phosphate [11, 22, 32]. The relationship between the levels of $CO_3^{2-}$ and $HCO_3^-$ and their adsorption ability for phosphate was observed. When the quantities of $CO_3^{2-}$ and $HCO_3^-$ were increased from 0.01 to 0.10 mmol·$L^-$

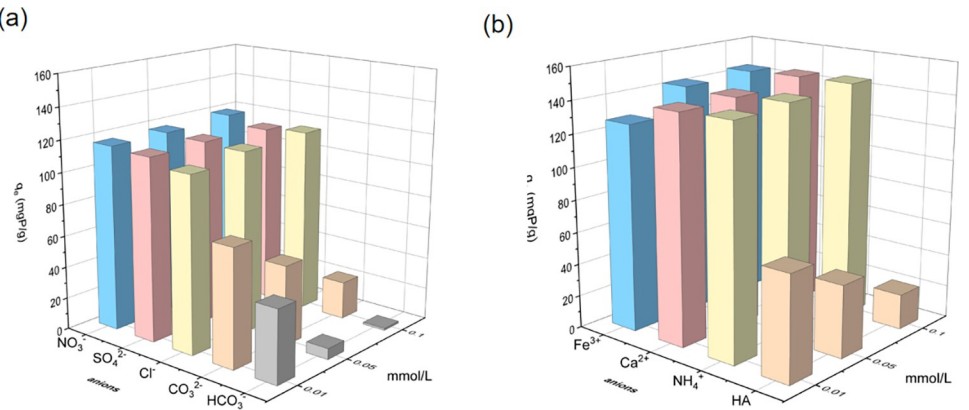

**Fig 8.** (a) The effect of coexisting anions on phosphate capture and (b) the effect of coexisting cations and gumic acids on phosphate capture.

[1], the adsorption capacity exhibited a drop from 77.76 mg·g$^{-1}$ to 23.66 mg·g$^{-1}$ for carbonate ions and from 45.48 mg·g$^{-1}$ to 1.23 mg·g$^{-1}$ for bicarbonate ions. The negative effect of $CO_3^{2-}$ and $HCO_3^-$ was due to their reaction with Mg (II), and the formation of precipitates reduced the adsorption of phosphate active sites by MgSBC-0.1. The adsorption capacity of MgSBC-0.1 for phosphate was diminished in the presence of HA. This could be attributed to the competitive interaction between functional groups of HA, such as carboxylic acid, carbonyl group, and phenolic hydroxyl group, and P for binding receptors on MgO and other oxides. Consequently, the adsorption capacity experienced a decline [35].

The promotional effect of the three cations on the adsorption of phosphate by MgSBC-0.1 is evident from the observation of Fig 8B, as compared to the control group. The adsorption capacity of MgSBC-0.1 for phosphate ions exhibited a steady increase as the ion concentration increased. For $NH_4^+$ ions, the dissolved MgO and $Mg(OH)_2$ on the surface of MgSBC-0.1 may precipitate with $NH_4^+$ to promote phosphate adsorption. Therefore, in water with a pH of approximately 7, the following reactions might occur on the surface of the composite material [36].

$$MgO + H_2O \rightarrow Mg(OH)_2 \rightarrow Mg^{2+} + 2OH^- \tag{9}$$

$$Mg^{2+} + HPO_4^{2-} + NH_4^+ + 6H_2O \rightarrow MgNH_4PO_4 \cdot 6H_2O(s) \tag{10}$$

The presence of $Ca^{2+}$ and $Fe^{3+}$ ions was found to enhance the adsorption of phosphate by MgSBC-0.1. This could be attributed to the involvement of Ca (particularly free CaO and $CaSO_4$) and iron (specifically $Fe_2O_3$) as the primary components responsible for phosphate fixation. The phosphate fixation of Ca components was formed by precipitation of Ca phosphate to form $CaHPO_4·2H_2O$, while the phosphate fixation of iron components was formed by ligand exchange or electrostatic attraction to form a coprecipitation of phosphate ions and iron oxide [37]. XPS characterization showed that MgSBC-0.1 itself contains Ca and Fe, which was conducive to the removal of phosphate in wastewater [38].

To investigate the efficacy of MgSBC-0.1 in removing phosphorus from real wastewater, adsorption tests were performed utilizing rural biogas slurry as the source water. The total phosphorus concentration and pH value of the biogas slurry were 26.92±2.00·mgP·L$^{-1}$, and 8.82±0.01·mgP·L$^{-1}$, respectively. Subsequently, upon increasing the dosage of MgSBC-0.1 from 0.33 g·L$^{-1}$ to 4.00·g·L$^{-1}$, there was observed an ascending pattern in the removal rate of total phosphorus inside the biogas slurry. This rate exhibited an increment from 33.38% to 87.38%. But for SBC, the removal rate of phosphate only increased from 18.58% to 52.71%, much lower than MgSBC-0.1 (S3 Fig). The residual concentrations of total phosphorus in the solution decreased from 6.78 to 1.82·mgP·L$^{-1}$. Upon reaching a concentration of 2.00·g·L$^{-1}$, the adsorbent demonstrated a stable removal rate of approximately 87.20%. This indicated that for MgSBC-0.1, 2.00·g·L$^{-1}$ was the optimal solid-liquid ratio for treating actual biogas slurry, and biochar modified with magnesium loading had better treatment effect on actual biogas slurry.

## Mechanism of P capture

The primary constituents of the biochar derived from sludge, as investigated in this work, consisted predominantly of Mg $(OH)_2$ and MgO. Additionally, the sludge material itself was found to contain the element Ca. Hence, the P and adsorption mechanism exhibited distinct characteristics in comparison to other investigations.

The XRD pattern (Fig 9A) shows that the MgO characteristic peak of MgSBC-0.1 was also weakened and the $MgHPO_4·3H_2O$ characteristic peak appeared after adsorption of phosphate, which also indicated the occurrence of the reaction. To gain deeper insights into the

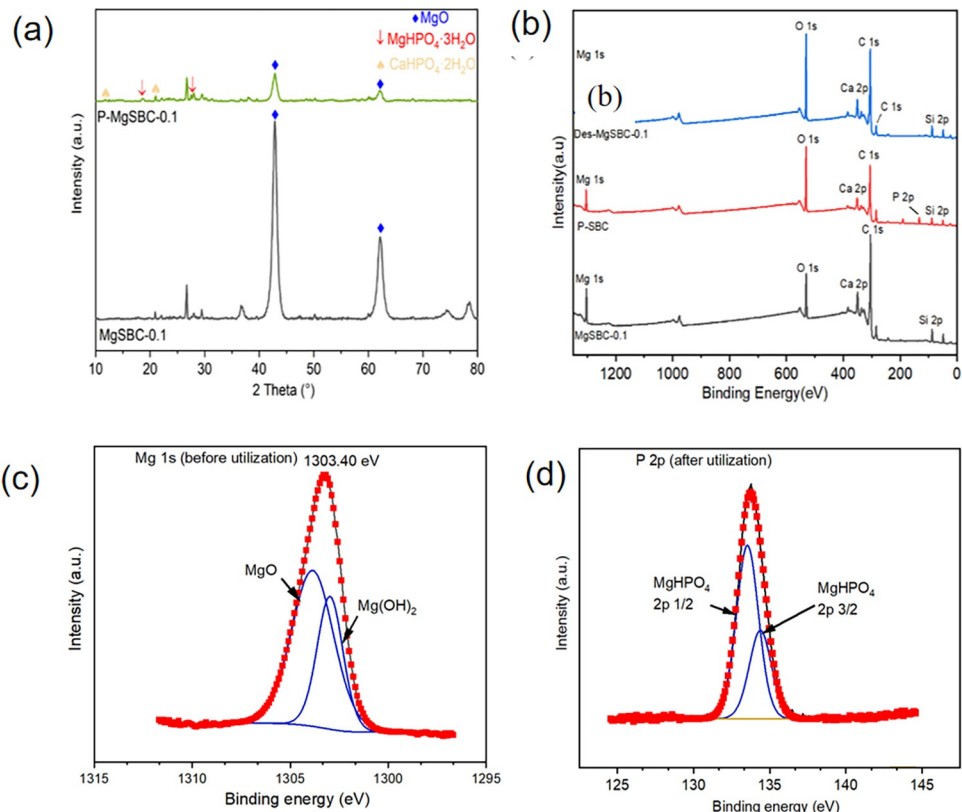

**Fig 9.** (a) XRD spectrum of P-MgSBC-0.1 biochar, (b) XPS spectra of MgSBC-0.1, P-SBC, and P-MgSBC-0.1, (c) XPS spectra of Mg 1s before utilization and (d) after utilization phosphate adsorption of MgSBC-0.1, respectively.

adsorption mechanism, the XPS spectrum revealed a substantial reduction in the P content in MgSBC-0.1, as indicated in Table 1. Furthermore, a substantial quantity of P was seen following the adsorption of phosphate by MgSBC-0.1. This finding further supported the notion that MgSBC-0.1 had a high affinity for phosphate capture. The XRD spectra of biochar (loaded with P) revealed prominent indications of pre-existing MgO. Additionally, the spectra exhibited the presence of newly formed Mg-P and Ca-P crystals, namely in the form of $MgHPO_4$, and $CaHPO_4$ (Fig 9A). In conjunction with the previously documented surface deposition mechanism of Mg-loaded biochar on P. It was worth noting that the Ca released from sludge-based biochar might also exert a substantial influence on the precipitation of phosphorus from aqueous solutions [39].

The confirmation of P precipitation on biochar was additionally verified via XPS analysis, as depicted in Fig 9B. The spectrum of Mg 1s (Fig 9C) revealed the existence of MgO and Mg (OH)$_2$ compounds on the surface of phosphorus-loaded biochar prior to adsorption. Conversely, the spectrum of P 2p (Fig 9D) demonstrated the presence of $MgHPO_4$ compounds on the surface of P-loaded biochar following adsorption. Despite the low solubility of both MgO and $Mg(OH)_2$, the introduction of P anions in the solution potentially enhanced their dissolution, resulting in the formation of more insoluble P salt, such as $MgHPO_4$.

The pH values of the solutions were measured prior to and after adsorption. It was observed that the initial pH value for all solutions was approximately 5.70, while the final pH value was around 7.40. These findings align with the precipitation mechanism. In the specified pH range, the phosphate present in the solution was shown to exist as $HPO_4^-$ and $H_2PO_4^-$ ions.

Consequently, the released Mg and Ca ions underwent precipitation, resulting in the formation of $MgHPO_4$ and $CaHPO_4$ crystals. In the presence of acidic or neutral conditions, the formation of outer surface complex equations between $HPO_4^{2-}$ and Mg and Ca surfaces occurred when OH- groups were present on the material's surface. The equations could be expressed as follows [40, 41],

$$MgO + H_2O \rightarrow Mg^{2+} + 2OH^- \tag{11}$$

$$MgO + H_2O \rightarrow MgOH^+ + OH^- \tag{12}$$

$$MgOH^+ + HPO_4^{2-} \rightarrow MgOH^+ \cdots HPO_4^{2-} \tag{13}$$

$$Mg^{2+} + HPO_4^{2-} \rightarrow MgHPO_4 \tag{14}$$

$$Ca-OH + H^+ \rightarrow= Ca-OH_2^+ \tag{15}$$

$$Ca-OH_2^+ + H_2PO_4^-/HPO_4^{2-} \rightarrow Ca-OH_2^+ ---H_2PO_4^-/HPO_4^{2-} \tag{16}$$

## Recovery of phosphate

To explore the desorption mechanism of phosphoric acid, Des-MgSBC-0.1 was characterized using XPS (Fig 9B) and FTIR (S4 Fig). The XPS spectrum showed that the P 2P peak 360 disappears substantially after alkali immersion, indicating phosphate desorption. The P 2P peak disappeared, and the binding energy of Mg 1s peak returns to a level equivalent to MgSBC-0.1, indicating the chemical bond cleavage between $Mg^{2+}$ and $HPO_4^{2-}$. And in the FTIR spectrum, the P-O characteristic peak at around 577 $cm^{-1}$ after desorption almost disappeared. The above results indicated that the alkaline solution could quickly desorb the phosphorus adsorbed on MgSBC-0.1.

The reusability of MgSBC-0.1 is shown in S5 Fig. The adsorption efficiency of MgSBC-0.1 for phosphorus exhibited a substantial decline as the number of adsorption desorption cycles increased. Specifically, the efficiency declined from 92.52% in the initial cycle to 69.22% in the subsequent cycle, and further decreased to 59.48% after the third cycle. The results indicated that the adsorption and reusability of MgSBC-0.1 for phosphorus was limited.

## Conclusion

MgSBC-0.1 was prepared by impregnating sludge powder with Mg acetate and pyrolysis at 500˚C. The increase in specific surface area of MgSBC-0.1 and the Mg-loading-modification enhance its ability to recover phosphate from aqueous solutions. The adsorption of phosphate by MgSBC-0.1 follows the Langmuir isotherm and pseudo second order kinetic model, with a maximum adsorption capacity of 379.52 mg·$g^{-1}$, which is 7.17 times higher that of SBC. In addition, common anions ($SO_4^{2-}$, $NO_3^-$, and $Cl^-$) have no substantial inhibitory effect on the adsorption of phosphate ions by MgSBC-0.1, while $CO_3^{2-}$ and $HCO_3^-$ have a substantial inhibitory effect. The easily generated cations ($Fe^{3+}$, $NH_4^+$, and $Ca^{2+}$) in wastewater all promote the adsorption of phosphate by MgSBC-0.1. The method of spontaneous capture of phosphate involves traditional three-step hydrolysis mechanisms, particularly electrostatic attraction and chemical precipitation. During this process, the adsorbed P mainly exists in the form of $MgHPO_4$ and $CaHPO_4$. The improvement of adsorption capacity is fundamentally due to the transformation of the main adsorption form from physical adsorption to chemical adsorption.

## Supporting information

**S1 Fig. Fitting of phosphate adsorption isotherms for SBC at 318K.**
(TIF)

**S2 Fig.** (a) Kinetic curves of SBC and (b) particle diffusion model fitting of SBC.
(TIF)

**S3 Fig. The removal effect of MgSBC-0.1 and SBC on actual biogas slurry.**
(TIF)

**S4 Fig. FTIR spectra of P-SBC and P-MgSBC-0.1.**
(TIF)

**S5 Fig. MgSBC-0.1 cycle regeneration.**
(TIF)

**S1 Graphical abstract.**
(TIF)

**S1 Data set.**
(ZIP)

## Author Contributions

**Conceptualization:** Chu-Ya Wang.

**Data curation:** Qi Wang.

**Formal analysis:** Qi Wang.

**Funding acquisition:** Chu-Ya Wang, Guangcan Zhu.

**Investigation:** Qi Wang.

**Methodology:** Heng-Deng Zhou.

**Software:** Xin Fang.

**Validation:** Qi Zeng, Guangcan Zhu.

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
