## [Decision Letter · Decision Letter 0]

2 Jan 2024

PONE-D-23-38741High-efficient adsorption of phosphate over a novel magnesium-loaded sludge-based biocharPLOS ONE

Dear Dr. Wang,

Thank you for submitting your manuscript to PLOS ONE. After careful consideration, we feel that it has merit but does not fully meet PLOS ONE’s publication criteria as it currently stands. Therefore, we invite you to submit a revised version of the manuscript that addresses the points raised during the review process.

We look forward to receiving your revised manuscript.

Kind regards,

Jorge Paz-Ferreiro, Ph.D.

Academic Editor

PLOS ONE

Journal Requirements:

"This work was supported by the Natural Science Foundation of Jiangsu Province

(BK20211047 and BK20220038)."

"This study was supported by Chu-Ya Wang from the Natural Science Foundation of Jiangsu Province (BK20211047).

This study was supported by Guangcan Zhu from the Natural Science Foundation of Jiangsu Province(BK20220038)."

"This study was supported by Chu-Ya Wang from the Natural Science Foundation of Jiangsu Province (BK20211047).

This study was supported by Guangcan Zhu from the Natural Science Foundation of Jiangsu Province(BK20220038)." 

Reviewers' comments:

Reviewer's Responses to Questions

**Comments to the Author**

1. Is the manuscript technically sound, and do the data support the conclusions?

Reviewer #1: Yes

Reviewer #2: Yes

Reviewer #3: No

Reviewer #4: Yes

2. Has the statistical analysis been performed appropriately and rigorously? 

Reviewer #1: Yes

Reviewer #2: Yes

Reviewer #3: No

Reviewer #4: N/A

3. Have the authors made all data underlying the findings in their manuscript fully available?

Reviewer #1: Yes

Reviewer #2: No

Reviewer #3: No

Reviewer #4: Yes

4. Is the manuscript presented in an intelligible fashion and written in standard English?

Reviewer #1: Yes

Reviewer #2: Yes

Reviewer #3: No

Reviewer #4: Yes

5. Review Comments to the Author

Reviewer #1: In this manuscript title "High-efficient adsorption of phosphate over a novel magnesium-loaded sludge-based biochar", writers studied adsorption of phosphate by novel magnesium-loaded sludge-based biochar comprehensively. A comprehensive study has been presented in an understandable language. Congratulations

Reviewer #2: This study investigates the removal of phosphate from aqueous solutions by synthesizing sludge-based biochar modified with Mg-loading-modification. Compared with unmodified sludge-based biochar, the synthesized MgSBC-0.1 has a 5.57-fold increase in specific surface area and can effectively remove phosphate within the initial solution pH range of 3.0 to 7.0. The maximum phosphorus adsorption capacity is 379.52 mg·g-1. The experimental design of the manuscript is comprehensive and rich in content. I suggest accepting it after correcting the following issues.

1.The data in the article is incomplete and lacks the figures S1, S2, S3, and 8 mentioned in the content.

2.In the preparation of MgSBC-0.1 mentioned in 2.1, the concentration of magnesium acetate soaking was not mentioned. What is the effect of different concentrations? It is also not specified what the target temperature is, is it the 500℃ mentioned below?

3.The description of the pores of the four types of sludge-based biochar in section 3.1 is not very clear, and there is little correlation between the data and conclusions. For example, I don't quite understand the analysis of the conclusion that "the pores of the impregnating material will be blocked, and through calcination, the pores will be expanded".

4.The Figure 4a mentioned at the beginning of 3.3 should be Figure 4b, as it is followed by a description of the relationship between phosphate adsorption capacity and pH.

5.In the last paragraph of 3.3, adsorption tests were performed utilizing rural biogas slurry as the source water to investigate the removal effect of MgSBC-0.1 on phosphorus in actual wastewater. Can a control group be set up here to indicate the current adsorption effect of unmodified sludge-based biochar or commonly used adsorbents?

6.Where is the XPS spectrum mentioned at the beginning of 3.5?

Reviewer #3: The present work studied the removal of phosphate (PO4-P) from an aqueous solution by synthesizing sludge-based biochar (MgSBC-0.1) from anaerobic fermentation sludge treated with Mg-loading-modification and compared it with unmodified sludge-based biochar (SBC). The authors have done a decent amount of experimental work, but the explanation is not up to the mark. There are so many grammatical and syntax errors, and in some places, sentences are very confusing and fail to explain the sentence's meaning. There are so many inconsistencies in analyzing the results; in various places, authors contradict their own claims. The following are the comments that must be addressed to improve the quality of work:

1. Some abbreviations are not defined in the manuscript (e.g., BC (biochar) and SP?). Crosscheck the whole manuicript for all such abbrivation errors.

2. There should be a gap between the numerical value and the unit.

3. Reframe the sentence in section 2.1. as “The sludge was dried in an oven at 105 °C to reduce water to less than 1% followed by ground to 200 mesh.”

4. At what temperature was the sludge powder calcined? It was not mentioned in the manuscript.

5. Reframe the sentence, “Initially, 30 mL of a 100·mg·L-1 phosphate solution was filled with the solution in a conical flask.”

6. In some places, authors used °C, and at some places used degrees. Make it °C throughout the manuscript.

7. Use a similar unit format throughout the manuscript. At some places, authors used inverse format, while at some places it's different (eg. degree per minute).

8. Round off the numerical values up to two decimal places.

9. There is a contradiction in the results between SEM and BET. The authors claimed that SP and SBC possessed particle-like structures covered with pores, with shallow pore structures although no pores can be seen in the fig. 1(a,b). And in the BET results they claimed that type II is for mesoporous structures, however, it is for nonporous or microporous adsorbents. Follow this article (https://doi.org/10.1351/pac200173020381) and write the proper explanation. Also add the TEM analysis to confirm more details. Why did the author report micropore volume when it was already confirmed that adsorbent has mesopores in nature? It is better to report mesopore volume.

10. In XRD analysis, authors mentioned four XRD patterns (SP, SBC, MgSBC-0.1 and P-MgSBC-0.1), however, there are only 3 XRD patterns in fig. 2b. Such type of mistakes are not acceptable, and they show the seriousness of work.

11. Make a single Figure for all the XRD spectra so that they can be compared easily, and rather than just mentioning the peaks, try to explain what changes occurred after the impregnation and how they help in adsorption.

12. In the XPS analysis, the authors claimed that N and Fe are present; however, these elements are not traced (as shown in Figure 2d). In Table 1, the sum of the percentage of all the elements is not equal to 100 for SP and MgSBC-0.1. Explain why this is so.

13. In FTIR analysis, the oxygen functional groups (C=O, C-O) show lesser intensity for MgSBC-0.1 in comparison to SBC and SP. Although XPS analysis confirmed that the oxygen percentage is increased, there is a contradiction between the results.

14. In section 3.2, the author claimed they tested four biochars. In the text, they mentioned MgSBC-0.01, but in the figure, they mentioned MgSBC-0.02.

15. After figure 4a they discussed the figure 5c and d. The discussion associated with the figure should be in proper order in the manuscript. You cannot describe any figure randomly.

16. The authors showed the XRD spectrum of MgBC600, I thought it would be abbreviated for 600°C calcination, and in conclusion, they reported the temperature is 500°C. It was still unclear to me at what temperatures they had done the calcination and why.

17. Refer to these papers for a more detailed explanation.

• Singh, V. and Srivastava, V.C., 2020. Self-engineered iron oxide nanoparticle incorporated on mesoporous biochar derived from textile mill sludge for the removal of an emerging pharmaceutical pollutant. Environmental Pollution, 259, p.113822.

• Singh, V., Chakravarthi, M.H. & Srivastava, V.C. Chemically modified biochar derived from effluent treatment plant sludge of a distillery for the removal of an emerging pollutant, tetracycline, from aqueous solution. Biomass Conv. Bioref. 11, 2735–2746 (2021). https://doi.org/10.1007/s13399-020-00683-4

Reviewer #4: The manuscript is well structured and written with minor errors and typos. My comments are:

• There is many literature prepared sludge-derived biochar. What is the novelty of your idea?

• You stated that the adsorption capacity could reach more than 700 mg/g with modified biochar but you get only maximum adsorption with 379.52 mg/g despite reaching high P concentrations 400 mg/L which isn't realistic in the natural environment. Please, clarify.

• I would recommend to amend the title and remove high-efficient.

• The following sentence is not clear. "Phasite equilibrium adsorption capacity with MgSBC-0.1" in 3.3 Effect of different factors on phosphate capture. Please, revise.

• You mentioned zetasizer in materials and method but there is no results or discussion about it.

• The last number of figures is 7 and you mentioned Fig. 8 in the mechanism section. Please, revise and oblige.

• The recovery of phosphate section is not completed. What are the quantitative amount of the phosphate recovered? What is the next step to do with recovered phosphate? You focused mainly on the reusability of MgSBC-0.1

• Conclusion section is long. Please, focus on the main findings and recommendations are missing.

6. PLOS authors have the option to publish the peer review history of their article (what does this mean?). If published, this will include your full peer review and any attached files.

Reviewer #1: **Yes: **Murat Yılmaz

Reviewer #2: No

Reviewer #3: No

Reviewer #4: **Yes: **Manal Fawzy

---

## [Author Response · Author response to Decision Letter 0]

11 Feb 2024

Response to Reviewer 1’s comments

In this manuscript title "High-efficient adsorption of phosphate over a novel magnesium-loaded sludge-based biochar", writers studied adsorption of phosphate by novel magnesium-loaded sludge-based biochar comprehensively. A comprehensive study has been presented in an understandable language. Congratulations

Thank you for your comments and congratulations.

 

Response to Reviewer 2’s comments

This study investigates the removal of phosphate from aqueous solutions by synthesizing sludge-based biochar modified with Mg-loading-modification. Compared with unmodified sludge-based biochar, the synthesized MgSBC-0.1 has a 5.57-fold increase in specific surface area and can effectively remove phosphate within the initial solution pH range of 3.0 to 7.0. The maximum phosphorus adsorption capacity is 379.52 mg·g-1. The experimental design of the manuscript is comprehensive and rich in content. I suggest accepting it after correcting the following issues.

Many thanks for the reviewer’s constructive suggestions. In response to your suggestions, we have carefully revised the manuscript and supplementary materials. Here is the response point by point.

1.The data in the article is incomplete and lacks the figures S1, S2, S3, and 8 mentioned in the content.

Thank you for your suggestion. We have corrected the error issue you raised regarding Figure 8 in this article. For Figures S1, S2, and S3, previously uploaded documents through the “Supporting Information" section”.

2. In the preparation of MgSBC-0.1 mentioned in 2.1, the concentration of magnesium acetate soaking was not mentioned. What is the effect of different concentrations? It is also not specified what the target temperature is, is it the 500℃ mentioned below?

Thanks for your suggestion. This section has been rewritten as follows:

“A 0.1 mol/L Mg(CH3COO)2 solution was prepared, from which a 100 mL sample was taken in a beaker, and 2 g sludge powder was added. The mixture was stirred at a constant rate at room temperature for 6 h, and then the beaker was placed in the oven for 24 h to remove any remaining water. Remove the dry material from the beaker and grind it until it passes through a 200 mesh sieve. The material after grinding was named MgSP, which was calcined in a tube furnace under a helium atmosphere at 500 ℃ for 2 h at a heating rate of 10 °C·min-1. After cooling, the material was removed and named MgSBC-0.1.”

3.The description of the pores of the four types of sludge-based biochar in section 3.1 is not very clear, and there is little correlation between the data and conclusions. For example, I don't quite understand the analysis of the conclusion that "the pores of the impregnating material will be blocked, and through calcination, the pores will be expanded".

Thank you for your question. We have revised this section to：

“In order to observe the changes in the pore structure of biochar after adding magnesium and calcination, SEM testing was conducted. The morphology of SP, SBC, MgSP-0.1, and MgSBC-0.1 is shown in the Fig. 1. SP and SBC have a granular structure covered by pores, with shallow pore structures (Fig. 1a and b). These pores and pore structures were irregular. The sludge powder impregnated with magnesium acetate has a smooth surface, slender pores, and a linear shape (Fig 1c). Through Mg-loading and calcination modification, the pores of MgSBC-0.1 are stretched, with some pore sizes increasing and pore depths deepening, providing additional surface area for adsorption and increasing adsorption sites (Fig. 1d).”

4. The Figure 4a mentioned at the beginning of 3.3 should be Figure 4b, as it is followed by a description of the relationship between phosphate adsorption capacity and pH.

Thank you for your suggestion. After verification, the corresponding error in Figure 4a mentioned at the beginning of 3.3 has been corrected.

5.In the last paragraph of 3.3, adsorption tests were performed utilizing rural biogas slurry as the source water to investigate the removal effect of MgSBC-0.1 on phosphorus in actual wastewater. Can a control group be set up here to indicate the current adsorption effect of unmodified sludge-based biochar or commonly used adsorbents?

Thank you for your suggestion. We have supplemented the experiment of SBC in the control group on the adsorption effect of actual biogas slurry, as shown in the following figure, and modified this paragraph as follows：

“To investigate the efficacy of MgSBC-0.1 in removing phosphorus from real wastewater, adsorption tests were performed utilizing rural biogas slurry as the source water. The total phosphorus concentration and pH value of the biogas slurry were 26.92±2.00·mgP·L-1, and 8.82±0.01·mgP·L-1, respectively. Subsequently, upon increasing the dosage of MgSBC-0.1 from 0.33 g·L-1 to 4.00·L-1, there was observed an ascending pattern in the removal rate of total phosphorus inside the biogas slurry. This rate exhibited an increment from 33.38% to 87.38%. But for SBC, the removal rate of phosphate only increased from 18.58% to 52.71%, much lower than MgSBC-0.1 (Fig.S3). The residual concentrations of total phosphorus in the solution decreased from 6.78 to 1.82·mgP·L-1. Upon reaching a concentration of 2.00·g·L-1, the adsorbent demonstrated a stable removal rate of approximately 87.20%. This indicated that for MgSBC-0.1, 2.00·g·L-1 was the optimal solid-liquid ratio for treating actual biogas slurry, and biochar modified with magnesium loading had better treatment effect on actual biogas slurry.”

Fig. S3 the removal effect of MgSBC-0.1 and SBC on actual biogas slurry.

6.Where is the XPS spectrum mentioned at the beginning of 3.5?

Thank you very much for your suggestion. The XPS spectrum mentioned at the beginning of 3.6 corresponds to Figure 9b. We have changed this paragraph to:

 "To explore the desorption mechanism of phosphoric acid, Des-MgSBC-0.1 was characterized using XPS (Fig. 9b) and FTIR (Fig. S4)."

The FTIR spectrum content and serial number of Figure have been updated.

Figure S4. FTIR spectra of P-SBC, P-MgSBC-0.1 and Des-MgSBC-0.1. 

Response to Reviewer 3’s comments

The present work studied the removal of phosphate (PO4-P) from an aqueous solution by synthesizing sludge-based biochar (MgSBC-0.1) from anaerobic fermentation sludge treated with Mg-loading-modification and compared it with unmodified sludge-based biochar (SBC). The authors have done a decent amount of experimental work, but the explanation is not up to the mark. There are so many grammatical and syntax errors, and in some places, sentences are very confusing and fail to explain the sentence's meaning. There are so many inconsistencies in analyzing the results; in various places, authors contradict their own claims. The following are the comments that must be addressed to improve the quality of work:

 Thank you for your suggestion. All the above issues have been well addressed in the revised draft. We have checked and corrected grammar and syntax errors. We have also reinterpreted the question you raised about the inability to explain the meaning of the sentence. Afterwards, corrections are made to the inconsistencies in the analysis results. Here is our response.

1. Some abbreviations are not defined in the manuscript (e.g., BC (biochar) and SP?). Crosscheck the whole manuicript for all such abbrivation errors.

We have accepted the suggestions of the reviewers and added definitions for SP and BC in the revised manuscript, as follows:

“The most typical way was to alter biochar (BC) with metal ions using various methods such as prediction.”

“A 0.1 mol·L-1 Mg (CH3COO)2 solution was prepared, from which a 100 mL sample was taken in a baker, and 2 g sludge powder (SP) was added.”

And we have also made modifications to other inappropriate abbreviations in the text, such as magnesium (Mg) - loading modification, etc.

2. There should be a gap between the numerical value and the unit.

Thank you for your suggestion. We have corrected any issues in the entire text.

3. Reframe the sentence in section 2.1. as “The sludge was dried in an oven at 105 °C to reduce water to less than 1% followed by ground to 200 mesh.”

Thank you for your suggestion. We have changed this paragraph to “The mixture was stirred at a constant rate at room temperature for 6 h, and then the beaker was placed in the oven for 24 h to remove remaining water. The dried material was removed from the beaker and then ground until it passed through a 200 mesh sieve.”

4. At what temperature was the sludge powder calcined? It was not mentioned in the manuscript.

Thank you for pointing out the issue. The sludge powder was calcined at 500 °C for 2 h. We have added this information to the material preparation section in section 2.1.

5. Reframe the sentence, “Initially, 30 mL of a 100·mg·L-1 phosphate solution was filled with the solution in a conical flask.”

Thanks for your suggestion. We rewrote the sentences in the revised manuscript as follow: 

“The dosage of adsorbent was fixed at 0.67 g·L-1.”

6. In some places, authors used °C, and at some places used degrees. Make it °C throughout the manuscript.

Thank you for your suggestion. The unit of temperature was unified as °C.

7. Use a similar unit format throughout the manuscript. At some places, authors used inverse format, while at some places it's different (eg. degree per minute).

Thank you for your suggestion. We have standardized all unit formats (e.g. °C·min-1) in the manuscript.

8. Round off the numerical values up to two decimal places.

Thank you for your suggestion. We have kept all numerical values in the manuscript to two decimal places

9. There is a contradiction in the results between SEM and BET. The authors claimed that SP and SBC possessed particle-like structures covered with pores, with shallow pore structures although no pores can be seen in the fig. 1(a,b). And in the BET results they claimed that type II is for mesoporous structures, however, it is for nonporous or microporous adsorbents. Follow this article (https://doi.org/10.1351/pac200173020381) and write the proper explanation. Also add the TEM analysis to confirm more details. Why did the author report micropore volume when it was already confirmed that adsorbent has mesopores in nature? It is better to report mesopore volume.

Thank you for your question. We have divided the questions to answer:

(1): There is a contradiction in the results between SEM and BET. The authors claimed that SP and SBC probable particle like structures covered with pores, with shared pore structures although no pores can be seen in Fig. 1 (a, b)

From the graph and combined with BET analysis, it can be concluded that the average pore sizes of SP, SBC, and MgSBC-0.1 are 28.52, 7.78, and 6.31 nm, with pore sizes ranging from 2-50 nm. According to the original text, "During the modification process, the average pore sizes of SP, SBC, and MgSBC-0.1 are still between 2-50 nm, all belonging to mesopores." This statement is correct.

Figure (a) (b) and (c): incremental pore volume vs. pore width images for SP, SBC, and MgSBC-0.1.

(2): In the BET results they claimed that type II is for mesoporous structures, however, it is for non-profit or microbial advisors

In the BET results, we reported that SP was a type II isothermal adsorption line, while SBC and MgSBC-0.1 are type IV adsorption isotherms For the N2 adsorption curve of SP, combined with the pore size distribution image in the above figure, we found that it also produced a hydrogen line, which was the same as the adsorption curve of MgSBC-0.1 and belongs to type IV adsorption isotherms We have made the following changes to the relevant content in the manuscript:

"According to the IUPAC classification, the N2 adsorption curves of SP, SBC, and MgSBC-0.1 showed type IV adsorption isotherms, accompanied by H3 hysteresis loops"

(3): Why did the author report micropore volume when it was already confirmed that adsorbent has mesopores in nature?

In the manuscript, we proposed the micropore volume of the material, but later on, we all proposed a micropore volume of 0 cm3·g-1 In order to avoid reader confusion, we have removed the micropore volume of 0 cm3·g-1 and added the mesoporous volume This paragraph is modified as follows:

"In terms of BET surface area (cm2·g-1) and pore volume (cm3·g-1), SBC had values of 17.76 and 0.037, while MgSBC-0.1 had values of 93.35 and 0.17."

In the revised manuscript, we have added TEM images of SP, SBC, and MgSBC-0.1, as well as mapping images of MgSBC-0.1, and added a paragraph as follows:

“The micro morphology of SP, SBC, MgSBC-0.1 is visualized by TEM images (Fig. 3a-d). SP and SBC have no obvious pore structure at 500 nm, but the morphology of mgsbc-0.1 was different from this. It can be clearly seen from Fig. 3d that MgSBC-0.1 produces more pores, indicating that more active sites may be exposed on the surface. The HRTEM image at 5 nm shows the d spacing between crystal planes of 0.209 nm (Fig. 2d), corresponding to (200) of MgO, which may indicate that MgO is loaded on the surface of biochar, but it needs further verification. The element distribution in the mapping diagram of EDS further confirmed the element distribution in MgSBC-0.1. As shown in Fig. 3e-j, Mg, O and Si elements are well dispersed in the whole particle.”

Figure. (a-c) TEM images (500nm) of SP, SBC and MgSBC-0.1

(d-e) TEM images of MgSBC-0.1

(f-i) Elemental mapping of MgSBC-0.1

10. In XRD analysis, authors mentioned four XRD patterns (SP, SBC, MgSBC-0.1 and P-MgSBC-0.1), however, there are only 3 XRD patterns in fig. 2b. Such type of mistakes are not acceptable, and they show the seriousness of work.

Thank you for your question. After verification, only the XRD images of SP, SBC, and MgSBC-0.1 are shown in Figure 2. The XRD images of P-MgSBC-0.1 are further analyzed in Figure 9a. This is a error in the original text, and this paragraph has been revised to:：

“It could be seen that SP was mainly composed of silicon dioxide (SiO2).”

11. Make a single Figure for all the XRD spectra so that they can be compared easily, and rather than just mentioning the peaks, try to explain what changes occurred after the impregnation and how they help in adsorption.

Thank you for your suggestion. We have merged the XRD spectra of SP, SBC, and MgSBC-0.1, as well as the PDF cards of MgO and Si2O, into one image, as shown in the following figure. The XRD analysis in this section has been modified in the revised draft as follows: 

“Fig. 2b shows the XRD patterns of the crystal structures of SP, SBC, and MgSBC-0.1. The results indicated that the peaks corresponding to the crystals present in biochar were matched with SiO2 (PDF # 85-0798) and MgO (PDF # 45-0946). The diffraction peaks obtained at 2θ= 36.86 ° (111), 42.83 ° (200), 62.30 ° (220), 74.69 ° (311), and 78.63 ° (222) match MgO, while the diffraction peaks obtained at 2θ= 20.86 ° (100), 26.64 ° (011), 36.54 ° (110), 39.47 ° (102), and 50.14 ° (11-2) match SiO2. It could be seen that SP was mainly composed of silicon dioxide (SiO2). Calcination increased the peak strength of SiO2 in biochar. In addition, the peak intensity of SiO2 in MgSBC-0.1 was lower than that of unmodified biochar. However, a MgO diffraction peak appeared in the XRD spectrum of MgSBC-0.1, with the strongest peak occurring at 2θ= 42.83°, indicating the presence of a large amount of MgO in the prepared biochar after impregnation and calcination, which was also confirmed by other techniques (XPS and TEM) [22] 

Fig. XRD patterns of SP, SBC and MgSBC-0.1

The XRD spectrum of P-MgSBC-0.1 had been analyzed in the mechanism analysis section later and was not included in this figure. Therefore, your suggestion on how to contribute to the adsorption of phosphate had been further explained in the following text.

12. In the XPS analysis, the authors claimed that N and Fe are present; however, these elements are not traced (as shown in Figure 2d). In Table 1, the sum of the percentage of all the elements is not equal to 100 for SP and MgSBC-0.1. Explain why this is so.

Thank you for your question. In XPS analysis, the proportion of N and Fe atoms was very low. For example, in MgSBC-0.1, they account for 2.57% and 0.17% respectively. Their peaks were not obvious and have little effect on the analysis of the phosphate adsorption mechanism of MgSBC-0.1. Therefore, they had not been further marked in the figure.

We have recalculated SP, SBC, and MgSBC-0.1, and corrected the atomic proportions as follows:

Table 1.

Sample C O Si Mg N Ca Fe O/C

SP 59.02 29.81 3.65 / 5.34 1.22 0.96 0.51

SBC 48.18 35.34 6.49 / 5.08 2.55 2.36 0.73

MgSBC-0.1 28.44 42.48 1.48 14.46 2.57 10.40 0.17 1.49

13. In FTIR analysis, the oxygen functional groups (C=O, C-O) show lesser intensity for MgSBC-0.1 in comparison to SBC and SP. Although XPS analysis confirmed that the oxygen percentage is increased, there is a contradiction between the results.

Thank you for your question. XPS analysis showed that the oxygen percentage of MgSBC-0.1 increases, but in FTIR, its oxygen functional groups (C=O, C-O) showed lower strength due to the formation of Mg-O, Mg- (OH) and other functional groups on the material, which were independent of C=O and C-O. Therefore, C=O and C-O might have lower strength than of SP, and the increase in oxygen percentage was partially caused by the metal oxygen functional groups

14. In section 3.2, the author claimed they tested four biochars. In the text, they mentioned MgSBC-0.01, but in the figure, they mentioned MgSBC-0.02.

Thank you for your question. After verification, it has been found that MgSBC-0.02 in the figure is a typo and is correct as MgSBC-0.01. It has been corrected in this figure.

Figure 4. The effect of magnesium loading on the adsorption of P by Mg-loaded sludge-biochar.

15. After figure 4a they discussed the figure 5c and d. The discussion associated with the figure should be in proper order in the manuscript. You cannot describe any figure randomly.

Accepting the reviewer’s suggestion, We have made revisions to the discussion of the figures in the manuscript and have arranged them in the appropriate order.

16. The authors showed the XRD spectrum of MgBC600, I thought it would be abbreviated for 600°C calcination, and in conclusion, they reported the temperature is 500°C. It was still unclear to me at what temperatures they had done the calcination and why.

Thank you for raising the question. MgSBC600 was the initial naming used for the material in the pre-experiment, but there was a mistake in synchronously modifying it. It had been changed to the correct name MgSBC-0.1. The calcination temperature of MgSBC-0.1 had been reinterpreted in the 2.1 Preparation of sludge biochar section. And using a heat treatment temperature of 500 °C was the initial pre-experiment conducted by calcining biochar at different temperatures. It was found that calcining biochar at 500 °C had the best adsorption effect on phosphate. However, due to the limited length of the article, it was not explained in the article.

17. Refer to these papers for a more detailed explanation.

• Singh, V. and Srivastava, V.C., 2020. Self-engineered iron oxide nanoparticle incorporated on mesoporous biochar derived from textile mill sludge for the removal of an emerging pharmaceutical pollutant. Environmental Pollution, 259, p.113822.

• Singh, V., Chakravarthi, M.H. & Srivastava, V.C. Chemically modified biochar derived from effluent treatment plant sludge of a distillery for the removal of an emerging pollutant, tetracycline, from aqueous solution. Biomass Conv. Bioref. 11, 2735–2746 (2021). https://doi.org/10.1007/s13399-020-00683-4

Thank you for your recommendation. We have carefully read these two references, corrected any errors in the manuscript, and have added them to the references section of the manuscript.

 

Response to Reviewer 4’s comments

The manuscript is well structured and written with minor errors and typos. My comments are:

• There is many literature prepared sludge-derived biochar. What is the novelty of your idea?

 Thank you for raising the question. Firstly, compared to the sludge-based biochar prepared in reported literature, the biochar by Mg-loading-modification in this study contained abundant Mg and Ca. Secondly, the presence of MgO, Mg (OH)2, and CaO on sludge-based biochar could cause strong chemical adsorption with phosphate, thereby increasing the adsorption capacity of phosphate. The adsorption capacity of phosphate was 7.17 times higher than that of SBC. The maximum adsorption capacity was 379.52 mg·g-1, which is much higher than other sludge-based biochar. Finally, after three cycles of adsorption desorption treatment, the material could still produce a removal rate of 59.48% for phosphate, demonstrating the potential reusability of modified sludge-based biochar.

• You stated that the adsorption capacity could reach more than 700 mg/g with modified biochar but you get only maximum adsorption with 379.52 mg/g despite reaching high P concentrations 400 mg/L which isn't realistic in the natural environment. Please, clarify.

Thank you for your question. After verification, the 700 mg/g phosphate adsorption capacity mentioned in the first section of the introduction was based on the article "Phosphorus removal from European water using modified biochar". The article suggested that an adsorption capacity of 600 mg/g or above was only achieved when the initial phosphate concentration was 1800 mg/L. However, in practical applications, such a high initial phosphate concentration was unlikely to occur. And when the initial concentration of phosphate was 400 mg/g, as shown in the figure below, the adsorption amount was about 150 mg/g, which was lower than the phosphate adsorption amount in this study.

• I would recommend to amend the title and remove high-efficient.

We have greatly appreciated the reviewer’s constructive comments. We have made the following changes to the title：

“Adsorption of phosphate over a novel magnesium-loaded sludge-based biochar”

• The following sentence is not clear. "Phasite equilibrium adsorption capacity with MgSBC-0.1" in 3.3 Effect of different factors on phosphate capture. Please, revise.

 Thanks for your suggestion. This sentence have been rewritten as follows:

“The composition of wastewater was complex, and the adsorption capacity of MgSBC-0.1 for phosphate was affected by various coexisting ions in the wastewater.”

• You mentioned zetasizer in materials and method but there is no results or discussion about it.

 Thank you for your question. We had planned to conduct a Zeta potential analysis experiment before, but when exploring the mechanism later, we found that MgSBC-0.1 had a small proportion of removing phosphate through electrostatic attraction. Additionally, considering other factors, we did not conduct this experiment and omitted its explanation in the materials and methods section. The modification had now been completed.

• The last number of figuresr is 7 and you mentioned Fig. 8 in the mechanism section. Please, revise and oblige.

 Thank you for your suggestion. This section has been modified.

• The recovery of phosphate section is not completed. What are the quantitative amount of the phosphate recovered? What is the next step to do with recovered phosphate? You focused mainly on the reusability of MgSBC-0.1

 After three adsorption desorption experiments, a 100mg/L phosphate solution could recover 59.48 mg·L-1. Many relevant research has shown that adsorption of phosphate saturated biochar is a promising soil amendment with safe, green, and sustainable characteristics. However, due to the focus of this article on improving phosphate adsorption performance and studying adsorption mechanisms, field experiments were not conducted, and only the possibility of using it for soil fertilization and remediation was proposed.

[1] Beatrice A, Varco JJ, Dygert A, Atsar FS, Solomon S, Thirumalai RVKG, et al. Lead immobilization in simulated polluted soil by Douglas fir biochar-supported phosphate. Chemosphere. 2022;292. doi: 10.1016/j.chemosphere.2021.133355. 

[2] Han Y, Choi B, Chen X. Adsorption and Desorption of Phosphorus in Biochar-Amended Black Soil as Affected by Freeze-Thaw Cycles in Northeast China. Sustainability. 2018;10(5). doi: 10.3390/su10051574.

[3] Li H, Wang Y, Zhao Y, Wang L, Feng J, Sun F. Efficient simultaneous phosphate and ammonia adsorption using magnesium-modified biochar beads and their recovery performance. Journal of Environmental Chemical Engineering. 2023;11(5). doi: 10.1016/j.jece.2023.110875.

[4] Yu J, Tang L, Pang Y, Zeng G, Wang J, Deng Y, et al. Magnetic nitrogen-doped sludge-derived biochar catalysts for persulfate activation: Internal electron transfer mechanism. Chemical Engineering Journal. 2019;364:146-59. doi: 10.1016/j.cej.2019.01.163.

[5] Zhang H, Chen C, Gray EM, Boyd SE, Yang H, Zhang D. Roles of biochar in improving phosphorus availability in soils: A phosphate adsorbent and a source of available phosphorus. Geoderma. 2016;276:1-6. doi: 10.1016/j.geoderma.2016.04.020.

• Conclusion section is long. Please, focus on the main findings and recommendations are missing.

Thanks for your suggestion. We rewrote conclusion section in the revised manuscript as follow:

“MgSBC-0.1 was prepared by impregnating sludge powder with Mg acetate and pyrolysis at 500 °C. The increase in specific surface area of MgSBC-0.1 and the Mg-loading-modification enhance its ability to recover phosphate from aqueous solutions. The adsorption of phosphate by MgSBC-0.1 follows the Langmuir isotherm and pseudo second order kinetic model, with a maximum adsorption capacity of 379.52 mg·g-1, which is 7.17 times higher that of SBC. In addition, common anions (SO42-, NO3-, and Cl-) have no significant inhibitory effect on the adsorption of phosphate ions by MgSBC-0.1, while CO32- and HCO3- have a significant inhibitory effect. The easily generated cations (Fe3+, NH4+, and Ca2+) in wastewater all promote the adsorption of phosphate by MgSBC-0.1. The method of spontaneous capture of phosphate involves traditional three-step hydrolysis mechanisms, particularly electrostatic attraction and chemical precipitation. During this process, the adsorbed P mainly exists in the form of MgHPO4 and CaHPO4. The improvement of adsorption capacity is fundamentally due to the transformation of the main adsorption form from physical adsorption to chemical adsorption.”

---

## [Decision Letter · Decision Letter 1]

17 Mar 2024

PONE-D-23-38741R1Adsorption of phosphate over a novel magnesium-loaded sludge-based biocharPLOS ONE

Dear Dr. Wang,

Thank you for submitting your manuscript to PLOS ONE. After careful consideration, we feel that it has merit but does not fully meet PLOS ONE’s publication criteria as it currently stands. Therefore, we invite you to submit a revised version of the manuscript that addresses the points raised during the review process.

We look forward to receiving your revised manuscript.

Kind regards,

Jorge Paz-Ferreiro, Ph.D.

Academic Editor

PLOS ONE

Journal Requirements:

Reviewers' comments:

Reviewer's Responses to Questions

**Comments to the Author**

1. If the authors have adequately addressed your comments raised in a previous round of review and you feel that this manuscript is now acceptable for publication, you may indicate that here to bypass the “Comments to the Author” section, enter your conflict of interest statement in the “Confidential to Editor” section, and submit your "Accept" recommendation.

Reviewer #2: All comments have been addressed

2. Is the manuscript technically sound, and do the data support the conclusions?

Reviewer #2: Yes

3. Has the statistical analysis been performed appropriately and rigorously? 

Reviewer #2: Yes

4. Have the authors made all data underlying the findings in their manuscript fully available?

Reviewer #2: Yes

5. Is the manuscript presented in an intelligible fashion and written in standard English?

Reviewer #2: Yes

6. Review Comments to the Author

Reviewer #2: 1. In the second paragraph of the Introduction, the sentence "The most typical way was to alter biochar (BC) with metal ions using various methods such as prediction" has a different font size from the surrounding text, and it also lacks punctuation at the end with a missing period. There are two instances of this type of error in the document.

2. The first sentence of the third paragraph in the Introduction states, "SBC is one of the best candidates for phosphorus adsorption in wastewater treatment plants, as it can low-cost obtain synthetic biochar from sewage sludge [1]." In this sentence, SBC, which refers to sludge-based biochar as mentioned earlier, is correctly identified as a top candidate for phosphorus adsorption in wastewater treatment plants. However, the clause following 'as' introduces ambiguity, as it implies SBC is a method that "can low-cost obtain synthetic biochar from sewage sludge," which could be confusing. It is suggested to revise this for clarity.

3. Equation (1) should include parentheses around C0-Ct to reflect the original intent of the formula, which is to calculate the change in concentration before determining the amount of phosphate adsorbed per unit of adsorbent, Qt. Moreover, the document does not adequately define Qt, only mentioning it as the amount of phosphate adsorbed, whereas it specifically refers to the amount of phosphate adsorbed per unit of adsorbent.

4.

In the first subsection of the "Results and Discussion" section titled "Characterization and morphology," the second paragraph states, "In terms of BET surface area (cm2·g-1) and pore volume (cm3·g-1), SBC had values of 17.76 and 0.037, while MgSBC-0.1 had values of 93.35 and 0.17, while MgSBC-0.1 had values of 93.35, 0.17, and 0.00." The last two sentences are repetitive.

5. In the section preceding "Mechanism of P capture," there is a sentence that reads, "Subsequently, upon increasing the dosage of MgSBC-0.1 from 0.33 g·L-1 to 4.00·L-1 there was observed an ascending pattern in the removal rate of total phosphorus inside the biogas slurry." There is a mistake in the units of measurement in the sentence.

7. PLOS authors have the option to publish the peer review history of their article (what does this mean?). If published, this will include your full peer review and any attached files.

Reviewer #2: No

---

## [Author Response · Author response to Decision Letter 1]

22 Mar 2024

Response to Reviewer 2’s comments

1. In the second paragraph of the Introduction, the sentence "The most typical way was to alter biochar (BC) with metal ions using various methods such as prediction" has a different font size from the surrounding text, and it also lacks punctuation at the end with a missing period. There are two instances of this type of error in the document.

Thank you for your suggestion. We have revised this sentence and other similar issues in the manuscript.

2. The first sentence of the third paragraph in the Introduction states, "SBC is one of the best candidates for phosphorus adsorption in wastewater treatment plants, as it can low-cost obtain synthetic biochar from sewage sludge [1]." In this sentence, SBC, which refers to sludge-based biochar as mentioned earlier, is correctly identified as a top candidate for phosphorus adsorption in wastewater treatment plants. However, the clause following 'as' introduces ambiguity, as it implies SBC is a method that "can low-cost obtain synthetic biochar from sewage sludge," which could be confusing. It is suggested to revise this for clarity.

Thank you for your suggestion. We have changed this sentence to:

“SBC has the advantage of low cost and is one of the best adsorbents for phosphorus treatment in wastewater treatment plants.”

3. Equation (1) should include parentheses around C0-Ct to reflect the original intent of the formula, which is to calculate the change in concentration before determining the amount of phosphate adsorbed per unit of adsorbent, Qt. Moreover, the document does not adequately define Qt, only mentioning it as the amount of phosphate adsorbed, whereas it specifically refers to the amount of phosphate adsorbed per unit of adsorbent. 

Thank you for your suggestion. However, it is not common for us to take (C0-Ct) as a formula alone after consulting the literature, so we only explain its meaning in the text, not as an additional formula. We have changed this paragraph to:

“Where Qt refers to the amount of phosphorus adsorbed by each unit of adsorbent at time t. Where C0 (mg·L-1) represents the initial phosphorus concentration, Ct (mg·L-1) represents the phosphorus concentration at time t, and (C0-Ct) refers to the change of phosphorus concentration.”

4.In the first subsection of the "Results and Discussion" section titled "Characterization and morphology," the second paragraph states, "In terms of BET surface area (cm2·g-1) and pore volume (cm3·g-1), SBC had values of 17.76 and 0.037, while MgSBC-0.1 had values of 93.35 and 0.17, while MgSBC-0.1 had values of 93.35, 0.17, and 0.00." The last two sentences are repetitive.

Thank you for your suggestion. This paragraph has been changed to:

“In terms of BET surface area (cm2·g-1) and pore volume (cm3·g-1), SBC had values of 17.76 and 0.037, while MgSBC-0.1 had values of 93.35 and 0.17.”

5. In the section preceding "Mechanism of P capture," there is a sentence that reads, "Subsequently, upon increasing the dosage of MgSBC-0.1 from 0.33 g·L-1 to 4.00·L-1 there was observed an ascending pattern in the removal rate of total phosphorus inside the biogas slurry." There is a mistake in the units of measurement in the sentence.

Thank you for your suggestion. We have checked and unified the unit of measurement of this sentence as “g·L-1”.

---

## [Editor Report · Decision Letter 2]

26 Mar 2024

Adsorption of phosphate over a novel magnesium-loaded sludge-based biochar

PONE-D-23-38741R2

Dear Dr. Wang,

We’re pleased to inform you that your manuscript has been judged scientifically suitable for publication and will be formally accepted for publication once it meets all outstanding technical requirements.

Kind regards,

Jorge Paz-Ferreiro, Ph.D.

Academic Editor

PLOS ONE